# Corrosion Behavior of Ultrafine-Grained CoCrFeMnNi High-Entropy Alloys Fabricated by High-Pressure Torsion

**DOI:** 10.3390/ma15031007

**Published:** 2022-01-27

**Authors:** Haruka Shimizu, Motohiro Yuasa, Hiroyuki Miyamoto, Kaveh Edalati

**Affiliations:** 1Graduate School of Mechanical Engineering, Doshisha University, Kyotanabe 610-0394, Japan; haruka1.shimizu@daikin.co.jp; 2Department of Mechanical Engineering, Doshisha University, Kyotanabe 610-0394, Japan; myuasa@mail.doshisha.ac.jp; 3WPI, International Institute for Carbon-Neutral Research (WPI-I2CNER), Kyushu University, Fukuoka 819-0395, Japan; Kaveh.edalati@kyudai.jp

**Keywords:** severe plastic deformation, high-pressure torsion (HPT), high-entropy alloys, ultrafine-grained materials (UFG), nanocrystalline materials, passivation, pitting corrosion

## Abstract

The influence of the nanocrystalline structure produced by severe plastic deformation (SPD) on the corrosion behavior of CoCrFeMnNi alloys with Cr contents ranging from 0 to 20 at.% was investigated in aqueous 0.5 M H_2_SO_4_ and 3.5% NaCl solutions. The resistance to general corrosion and pitting became higher in both the solutions, with higher passivation capability observed with increasing Cr content, and it is believed that the high corrosion resistance of CoCrFeMnNi alloys can be attributed to the incorporation of the Cr element. However, the impact of the nanocrystalline structure produced by SPD on the corrosion behavior was negligibly small. This is inconsistent with reports on nanocrystalline binary Fe–Cr alloys and stainless steels processed by SPD, where grain refinement by SPD results in higher corrosion resistance. The small change in the corrosion behavior with respect to grain refinement is discussed, based on the passivation process of Fe–Cr alloys and on the influence of the core effects of HEAs on the passivation process.

## 1. Introduction

As a well-known high-entropy alloys (HEA), equiatomic CoCrFeMnNi alloy, also known as Cantor alloy, has been increasingly investigated in recent years, with a focus on enhanced fracture toughness and ductility at cryogenic temperatures [1]. HEAs could become future materials for structural applications due to these superior properties. However, compared with the mechanical properties, the corrosion properties, which are also important for structural applications, have not been investigated to the same extent. The majority of HEAs, as represented by Cantor alloy, have a higher amount of Cr and Ni compared to commercial stainless steels (SS) such as AISI304, so that they are expected to exhibit high corrosion resistance, and they have been demonstrated to possess high passivating capability in neutral solutions [2,3,4,5,6,7]. However, it seems that the corrosion resistance of HEAs is not necessarily as high as expected given the large Cr and Ni content [2,3,8], and in particular the pitting corrosion resistance in chloride-containing environments is inferior to AISI304 [2,4,8].

Grain refinement to a nanocrystalline structure has a significant influence on the mechanical properties and corrosion behavior of metallic materials. This strong effect is associated with a very small grain size and a large number of structural defects, i.e., grain boundaries and triple points. Such a high fraction of structural defects in nanocrystalline materials can lead to a significant increase in stored energy, which may increase reactivity. This phenomenon is expected to have a dual effect on corrosion behavior, which is dependent upon the material/environment system, as reviewed in [9,10]. As a general rule, a nanocrystalline structure in passivating electrolytes exhibits an improvement in corrosion resistance, whereas in depassivating electrolytes it exhibits a decrease in corrosion resistance [9,10]. The effect of grain size on the corrosion behavior of HEAs has been little studied to date [11,12]. Wang et al. examined the effect of grain size on the corrosion behavior of CoCrFeMnNi alloys and found that both fine-grained (<1.24 μm) and coarse-grained samples (145 μm) exhibited lower corrosion resistance than medium-grain-size specimens [11]. Han et al. examined the corrosion behavior of ultrafine-grained CoCrFeMnNi alloys produced via cryo-rolling and subsequent annealing for a short time, and they found that ultrafine-grained alloys exhibited less corrosion resistance than their coarse-grained counterparts [12]. In these studies, grain size was controlled by plastic deformation and subsequent annealing, and post-deformation annealing may induce elemental segregation or precipitation of phases such as the σ-phase, which degrade corrosion resistance.

Severe plastic deformation (SPD) techniques enable the fabrication of ultrafine-grained (UFG) or nanocrystalline (NC) materials in bulk forms suitable for load-carrying structural applications [13]. With the advent of SPD, there has been an increasing demand to clarify the effect of UFG/NC formation on corrosion from a practical viewpoint, especially with regard to the classic topic of the effect of grain size on corrosion [14,15]. Among the various materials investigated to date, stainless steels exhibited mostly improved corrosion resistance by forming UFG structures by SPD [10,14,15,16,17,18,19,20]. The enhanced corrosion resistance of UFG/NC SSs compared with coarse-grained (CG) materials has been mostly attributed to greater Cr enrichment in the passivation film [19,20,21,22,23,24,25,26,27], which has been validated experimentally by X-ray photoelectron spectroscopy (XPS) [24] and recently by Cs-corrected scanning-transmission electron microscopy (STEM) [28,29]. Two possibilities for the enhanced Cr enrichment in the passivation film of UFG/NC materials have been proposed. One is the selective dissolution of Fe and oxidation of Cr due to the high chemical reactivity of the UFG structure [21,22], and the second is fast diffusion of Cr from the metal matrix to the passivation film through high-density grain boundaries in the UFG alloys [24,25,26,27]. In this context, Cr-containing HEAs, represented by equiatomic CoCrFeNiMn alloys, could be expected to exhibit higher corrosion resistance due to the UFG/NC structure.

Among the various SPD techniques, processing by high-pressure torsion (HPT), where a specimen disk is subjected to a high applied pressure and concurrent torsional strain, is especially effective for producing a microstructure with exceptionally small grain sizes and a large fraction of high-angle grain boundaries [30,31]. Nanocrystalline CoCrFeMnNi HEAs have been fabricated by HPT, and their mechanical properties [32,33,34], microstructures [32,35], thermal stability [32] and superplasticity [36,37] have been studied. However, there have been no reports that describe the influence of HPT processing on the corrosion behavior of HEAs.

The purpose of the present research was to clarify the effect of UFG formation by SPD on the corrosion behavior of high-entropy Cantor alloys with various Cr contents in H_2_SO_4_ and NaCl solutions.

## 2. Materials and Methods

CoCrFeMnNi HEAs with a systematically varied Cr content ranging from 0 to 20 at.% were prepared by arc melting. The chemical compositions of the samples are shown in Table 1. The other elements, i.e., Co, Fe, Mn and Ni, were adjusted to be equiatomic, ranging from 20 to 25 at.%. The as-cast alloys were annealed at 1273 K for 16 h for homogenization in an Ar atmosphere, and this was repeated four times. The alloys were then cut by electrical discharge machining into 10 mm diameter and 1.0 mm thick disks. The disks were subjected to HPT at room temperature with a pressure of 6.0 GPa at 1 rpm for a total of 10 turns. After the homogenization treatment (referred to as CG1-5) and HPT processing (HPT1-5), the samples were evaluated in terms of Vickers microhardness, microstructure, X-ray diffraction (XRD) characteristics and corrosion behavior. The XRD structural analysis was performed using Co Kα radiation (*λ* = 1.7902 A) with a scanning step of 0.01 deg and a scanning speed of 2 deg/min. Field-emission transmission electron microscopy (FE-TEM, JEM-2100F, JEOL), equipped with energy-dispersive X-ray spectroscopy (EDS) was used to observe the microstructure and elemental distribution after HPT.

Potentiodynamic polarization tests were conducted in 1 M H_2_SO_4_ and 3.5 wt.% NaCl solutions at room temperature, using a potentiostat (HZ-5000, Hokuto-Denko Co Ltd., Tokyo, Japan) at a scan rate of 20 mV/min with an Ag/AgCl reference electrode in a saturated KCl solution. Each sample was immersed in the solution for one hour prior to the polarization tests. Dissolved oxygen was removed from the solution with argon gas before corrosion testing.

To investigate the chemical composition of passivation films on the equiatomic CG5 and HPT5 samples (20 at.% Cr), the samples were charged under potentiostatic conditions for 2 h at a potential of +400 mV_Ag/AgCl_ in 1 M H_2_SO_4_ solution, to produce stable passivation films. The potential was chosen with reference to the characteristic features of the potentiodynamic polarization curve. Cross-sectional TEM samples of the passivation film formed on the alloys after corrosion tests were prepared using focused ion beam milling (FIB, NB5000, Hitachi Hi-technology, Tokyo, Japan) at an acceleration voltage of 10–40 kV, following the deposition of a Pt layer to protect the surface of the alloys. Cross-sectional images of the passivation films formed on the alloys were characterized using high-resolution transmission electron microscopy (HRTEM), and the chemical composition of the passivation films was investigated using Cs-corrected high-angle annular dark-field STEM (Cs-corrected HAADF-STEM) (JEM NEOARM, JEOL, Akishima, Tokyo, Japan) equipped with EDS (JED-2300T, JEOL, Akishima, Tokyo, Japan)) at an acceleration voltage of 200 kV.

## 3. Results

Figure 1 shows the XRD patterns for the initial homogenized samples (designated CG1 to 5) and HPT samples (20 at.% Cr designated as HPT5). They all exhibit a typical single face-centered cubic (fcc) phase, and peak broadening is recognized for the HPT5 samples. There is no evidence of the formation of a new phase or the occurrence of phase transformation during the HPT processing. This stable single fcc phase of the Cantor alloy after HPT processing was also reported by other researchers using XRD [33], and the alloy remained a true single-phase solid solution down to the atomic scale, as demonstrated using three-dimensional atom probe tomography (3D-APT) [32].

Figure 2 shows the distribution of microhardness for the CG5 and HPT1-5 samples along the diameter. The lower dashed line at Hv = 120 corresponds to the initial hardness for the homogenized condition of CG5. The hardness was almost completely uniform across the entire diameter after HPT for 10 turns, and compared to the homogenized condition, the hardness was increased significantly by a factor of 4 to Hv = 510 in all samples.

Shahmir et al. processed equiatomic CoCrFeNiMn alloys using HPT, and the materials reached a saturation hardness of Hv = 450, which is slightly smaller than the value obtained in the present study. They also reported that fully uniform hardness was not achieved even after 10 turns; a small center area was softer than the edge part [33]. The reason for this inconsistency with the present results is unknown; however, the initial grain size or the level of slippage between the anvil and the disc may be responsible.

Bright-field TEM images at the center and edge parts of HPT5 samples are shown in Figure 3. The microstructure consists of small grains with an average grain size of about 40 nm, many of which are surrounded by curved or ill-defined grain boundaries. Selected area electron diffraction (SAED) analysis revealed well-defined ring patterns, which indicates that the grain boundaries mostly had high angles of misorientation. Both the bright-field images and the SAED analysis indicate that there was essentially no distinct difference in the microstructure between the center and edges, as also indicated by the Vickers microhardness test. Elemental distributions of the HPT5 sample observed using STEM-EDS are shown in Figure 4. The distribution of all elements was fairly uniform, and there was no visible segregation or agglomeration after HPT.

Figure 5 shows the potentiodynamic polarization curves for all samples in 0.5 M H_2_SO_4_ solution. All the CG and HPT specimens passed through the active, active–passive transition and passive regions, followed by second active, passive and transpassive regions, with increasing anodic polarization potential. Both the peak current *i*_p_ at the active–passive transition and the steady-state passive current *i*_pass_ decreased with increasing Cr content in both the CG and HPT alloys, which indicates that Cr increases the passivation ability, as documented in binary Fe–Cr alloys [22]. Therefore, Cr has a predominantly strong influence on the corrosion behavior among the constituent elements of these HEAs. The secondary passivation is associated with the oxidation of MnO_2_, while the final breakdown is dominated by the dissolution of MnO_4_ [7]. Therefore, the secondary peak position was almost the same for all the alloys despite the different amounts of Cr. Figure 6 compares the polarization curves for the CG and HPT specimens with the same amount of Cr. For all Cr contents, the polarization curves almost completely overlapped for the HPT and CG samples, which indicates that nanocrystalline formation did not alter the electrochemical behavior, irrespective of the amount of Cr. The peak current density *i*_p_ of the active region and the passive current *i*_pass_ are shown as a function of the Cr content in Figure 7. Data for high-purity Fe–Cr alloys and AISI304SS are also shown for reference [20,22,38]. Note that the fitting lines for *i*_p_ and *i*_pass_ for both the present alloys and the binary Fe–Cr alloys are almost matched, except for *i*_pass_ for the Fe–Cr alloys reported by Gupta et al. [22]. This indicates that Cr exclusively influences the passivation behavior. According to the results of Gupta et al. [22], both *i*_p_ and *i*_pass_ in nanocrystalline structures are smaller than in their coarse-grained counterparts. Figure 8 shows the potentiodynamic polarization curves in 3.5%NaCl solutions. Both the HPT and CG alloys showed spontaneous passivity with increasing overpotential, and the passivation films prevented continuous dissolution of the surface. Current fluctuations were conspicuous in the entire passive region and are attributed to the formation and repassivation of metastable pits. While passive currents gradually increased with the anodic overpotential in the CG samples, those for the HPT samples stayed constant until the breakdown potential, which indicates higher protective properties of the passivation films of the HPT samples. The passivation breakdown potential due to pit formation, *E*_pit_, increases with Cr content in both the CG and HPT specimens. It was confirmed that the pits were not localized in the center part, which has an immature microstructure compared with the edge part. Figure 9 shows the critical pitting potential *E*_pit_ as a function of the pitting resistance equivalent number (PRE). PRE (= Cr + 3.3 Mo) provides an empirical relative estimation of the pitting corrosion resistance for Fe–Cr and Fe–Cr–Ni alloys based on experimental fitting of the pitting potentials when exposed to NaCl solution. It is apparent that the pitting potentials for the HEAs are on the same fitting line as for high-purity Fe–Cr alloys (Figure 9). These results indicate that the resistance to pitting of the HEAs is determined exclusively by Cr, and the effect of other elements such as Ni and Co on the passivity is limited. If one compares the pitting potential with those of 304 SS [39,40], the pitting potential of 304 SS is higher than that in the present results despite having the same PRE, and it increased with the formation of the UFG structure. However, little difference in *E*_pit_ was observed between the HPT and CG samples, which suggests that UFG formation by HPT has little impact on the protective properties of the passivation film and the corrosion resistance.

Figure 10 shows the EDS line profiles for the CG5 and HPT5 specimens after polarization in the passive region (400 mV_Ag/AgCl_) for 2 h in 0.5 M H_2_SO_4_ solution. Figure 11 shows the corresponding STEM images and elemental maps. The layer with a high O signal is considered to be the passivation film. A Cr-rich oxide layer was observed on both CG and HPT specimens, which indicates that a stable passivation film can form on both specimens and that Cr is the main passivation former. In binary Fe–Cr alloys, Cr enrichment of the passivation films, which has mainly been observed using XPS, is due to the selective dissolution of Fe and the higher oxidation tendency of Cr [41,42,43], and a higher Cr/(Fe + Cr) ratio in the passivation film results in higher corrosion resistance. In HEAs, higher dissolution of Fe, Ni, Co and Mn compared to Cr is considered to occur during anodic dissolution, and this could result in Cr enrichment. The levels of Cr enrichment, expressed as the Cr/(Fe + Cr + Co + Ni + Mn) ratio, are comparable in both CG and UFG specimens and are smaller than 0.5, and this is consistent with the similar corrosion behavior of the two samples. According to Gupta et al. [22], the level of Cr enrichment Cr/(Fe + Cr) of the passivation film in Fe–20%-Cr alloys passivated in H_2_SO_4_ solutions increases with NC formation and reaches 0.8 [22], which is higher than the present result. Therefore, it is considered that the selective dissolution of other elements during the anodic reaction is not facilitated in HEAs compared with that of Fe in Fe–Cr alloys. The Ni-rich phenomenon underneath the surface oxide film was also observed in CoCrFeMnNi alloy [5], and this is commonly seen in studies of austenitic stainless steels [29].

## 4. Discussion

In the studied CoCrFeMnNi alloys, the amount of Cr had a strong effect on the electrochemical behavior, and the passivity became more stable with increasing amounts of Cr. This higher pitting potential in NaCl solution and greater protective properties of passivation films in H_2_SO_4_ with a higher amount of Cr has been established in binary Fe–Cr alloys and stainless steels. However, in spite of extensive reports of enhanced corrosion resistance in UFG Fe–Cr alloys, in the present study, the effect of UFG formation on the corrosion resistance was less than expected; UFG formation via HPT of CoCrFeMnNi had little or no impact on the corrosion behavior.

Cr enrichment of passivation films in binary Fe–Cr alloys has often been attributed to the selective dissolution of Fe and the oxidation of Cr [41,42,43,44]. Higher corrosion resistance of UFG Fe–Cr alloys fabricated by SPD than that of CG alloys, which has been reported by many researchers [10,14,15,16,17,18,19,20], is mostly attributed to greater Cr enrichment of the passivation film [19,20,21,22,23,24,25,26,27]. This local enrichment of Cr in the passivation film was mostly validated experimentally using XPS and established to date. This greater Cr enrichment in UFG Fe–Cr alloys has been explained by the following two processes. The first explanation is accelerated selective dissolution of Fe and oxidation of Cr [21,22]. UFG materials have a high density of grain boundaries, and atoms at grain boundaries have higher chemical activity, which results in greater dissolution into the solution and oxidation. The second explanation is the faster diffusion of Cr from the base metal to the passivation film through high-density grain boundaries which act as fast diffusion channels in UFG alloys [24,25,26,27]. Significant diffusion of Cr in nanostructured Fe–Cr alloys was reported by Wang et al., where Cr diffusion in nanostructured Fe–Cr alloys was 7 to 9 times faster than in the Fe lattice, and 4 to 5 times faster than at the grain boundaries of Fe [45]. This fast diffusion in nanostructured Fe is attributed to a high density of non-equilibrium grain boundaries [45].

The present results for CoCrFeMnNi alloys on the stable passivation composition against UFG formation (or less sensitive composition to grain size than Fe–Cr alloys) can be explained by considering some high-entropy effects, which are among the four core effects of HEAs. The high-entropy effect of HEAs decreases the possibility of elemental segregation or local enrichment of specific elements and favors the formation of a uniform distribution of component elements. Therefore, the selective dissolution of specific elements, which leads to local enrichment of the other remaining elements, increases the mixing enthalpy and may destabilize the fcc phase. Thus, selective dissolution could be suppressed during anodic dissolution. Luo et al. conducted an in situ element-resolved corrosion analysis using online ICP-MS analysis and concluded that no obvious selective dissolution of Mn, Co, Ni and Fe occurred during surface passivation of the equiatomic HEA in 0.1 M H_2_SO_4_ [3]. The second reason for the small influence of grain size on the corrosion behavior of CrCoFeMnNi alloys compared with Fe–Cr alloys is related to diffusion. The kinetics of diffusion are considered to be low in HEAs [46], and the so-called sluggish diffusion suppresses Cr diffusion from the base metal to the passivation film and thus suppresses Cr enrichment, in spite of UFG formation with high-density grain boundaries. In addition, grain boundaries in HEAs may not provide a fast diffusion channel due to grain boundary segregation [47] and a low average grain boundary energy [48,49]. Since the grain boundary is regarded as surplus energy when a grain boundary is embedded in a single crystal, the grain boundary energy in HEAs is significantly reduced due to the severe lattice distortion inside the grains [48,49]. This is because the energy difference between the grain boundaries and grain interior becomes small. Therefore, grain refinement or a high density of grain boundaries may not facilitate Cr diffusion and local enrichment in the passivation film during anodic dissolution in CrCoFeMnNi alloys.

## 5. Conclusions

The corrosion behavior of UFG CoCrFeMnNi alloys processed by HPT with Cr contents ranging from 0 to 20 at.% was investigated in 3.5% NaCl and 0.5 M H_2_SO_4_ solutions and compared to homogenized coarse-grained (CG) counterpart materials. Several conclusions were obtained, as follows.

The polarization curves in 1 M H_2_SO_4_ solution showed the active–passive, passive and transpassive behavior typical of binary Fe–Cr alloys. An increase in Cr content decreased the potential of the active–passive transition, and thus increased the ability to passivate. Cr was the dominant element providing the passivation ability.

The polarization curves in 1 M H_2_SO_4_ solution showed little difference between the coarse-grained state after homogenization and the nanocrystalline structure after HPT. The impact of nanocrystalline structure on corrosion was negligibly small. Similarly, corrosion resistance to pitting in 3.5% NaCl solution was also little affected by the formation of the nanocrystalline structure, and this is in striking contrast to the high corrosion resistance of nanocrystalline or ultrafine-grained stainless steels processed by SPD reported elsewhere.

Moderate Cr enrichment in the passivation films was observed equally for coarse-grained and nanocrystalline structures. This stable, less-sensitive composition in the passivation film with respect to grain refinement may originate from the core effects, i.e., severe lattice distortion and sluggish diffusion of high-entropy alloys.

## Figures and Tables

**Figure 1 materials-15-01007-f001:**
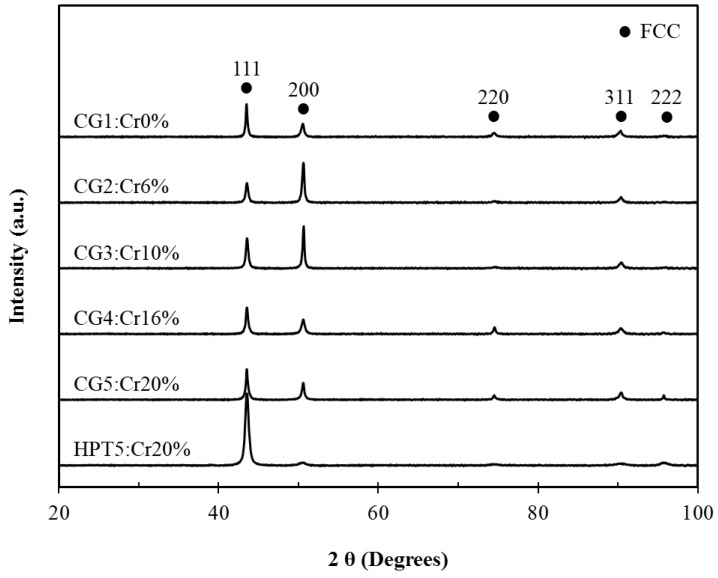
X-ray diffraction spectra after homogenization and HPT processing.

**Figure 2 materials-15-01007-f002:**
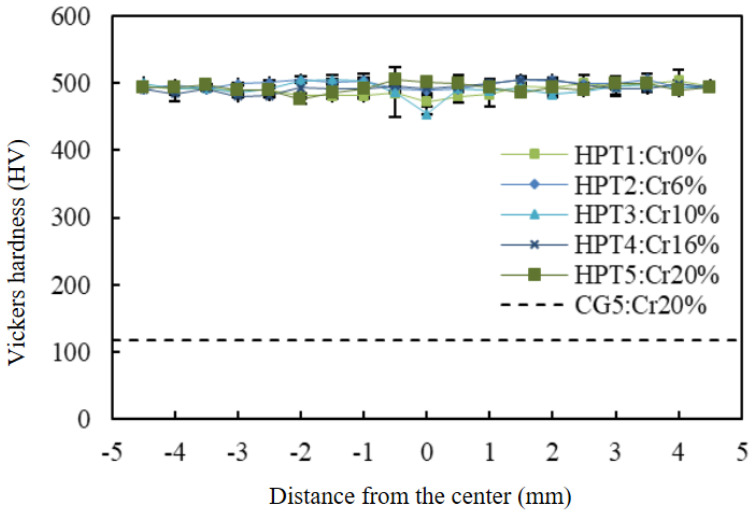
Vickers microhardness profiles after 10 turns of HPT for all samples.

**Figure 3 materials-15-01007-f003:**
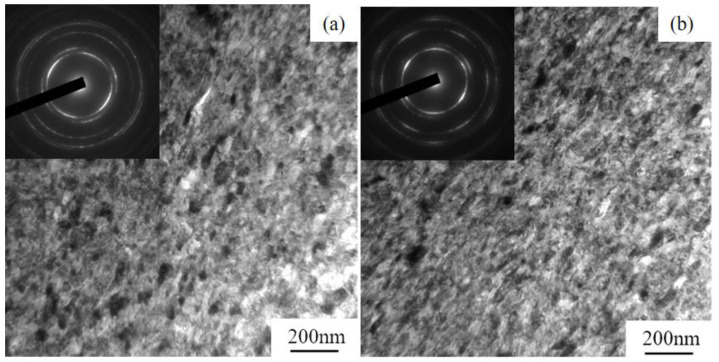
TEM images and corresponding SAED patterns of 20 at.% Cr after 10 turns of HPT (HPT5): (**a**) center; (**b**) edge part of the disc.

**Figure 4 materials-15-01007-f004:**
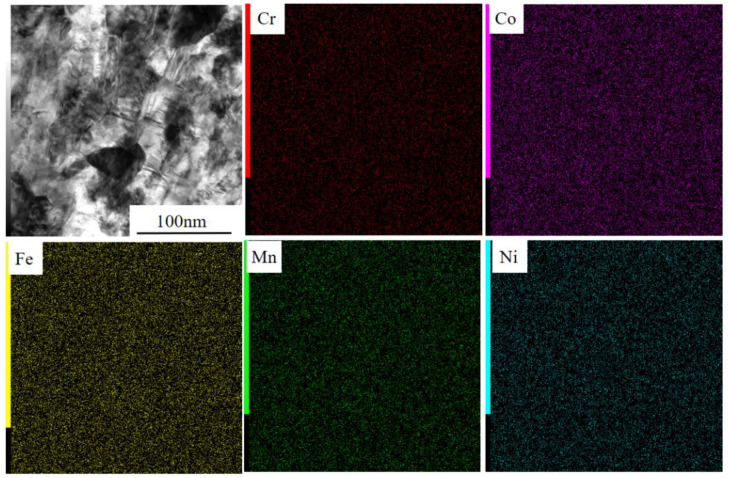
Elemental maps obtained by STEM-EDS for 20 at.% Cr after 10 turns of HPT (HPT5).

**Figure 5 materials-15-01007-f005:**
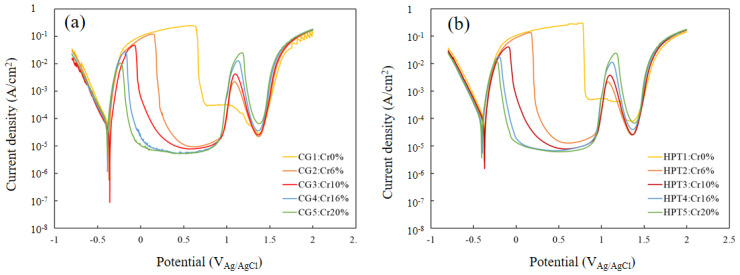
Dynamic polarization curves in 1 M H_2_SO_4_ solution after (**a**) homogenization (CG) and (**b**) HPT processing (HPT).

**Figure 6 materials-15-01007-f006:**
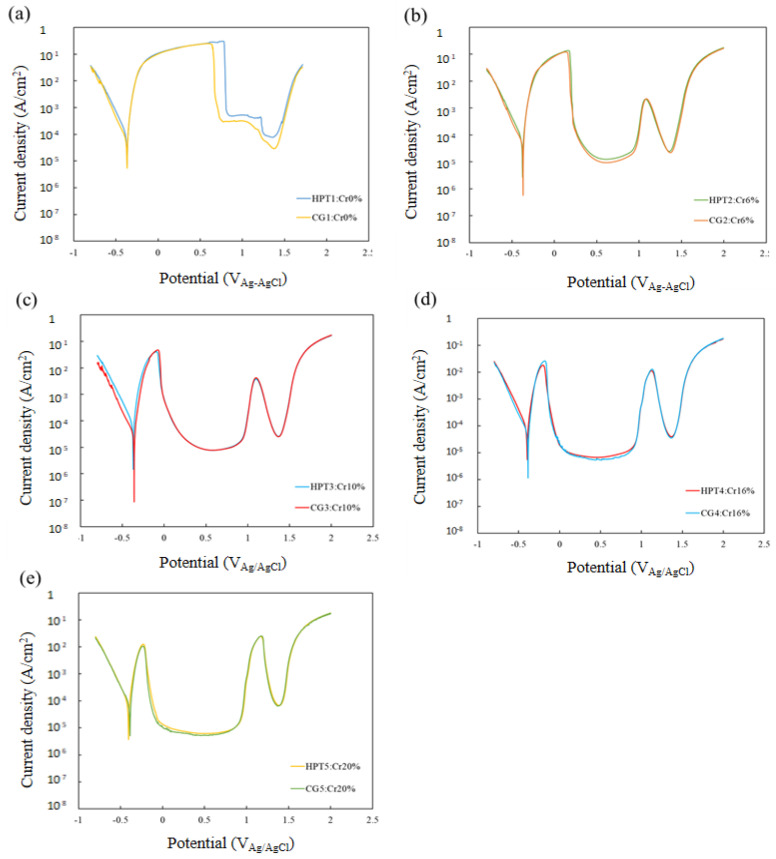
Comparison of dynamic polarization curves after homogenization (CG) and HPT processing (HPT) of (**a**) 0 at.% Cr, (**b**) 6 at.% Cr, (**c**) 10 at.% Cr, (**d**) 16 at.% Cr and (**e**) 20 at.% Cr.

**Figure 7 materials-15-01007-f007:**
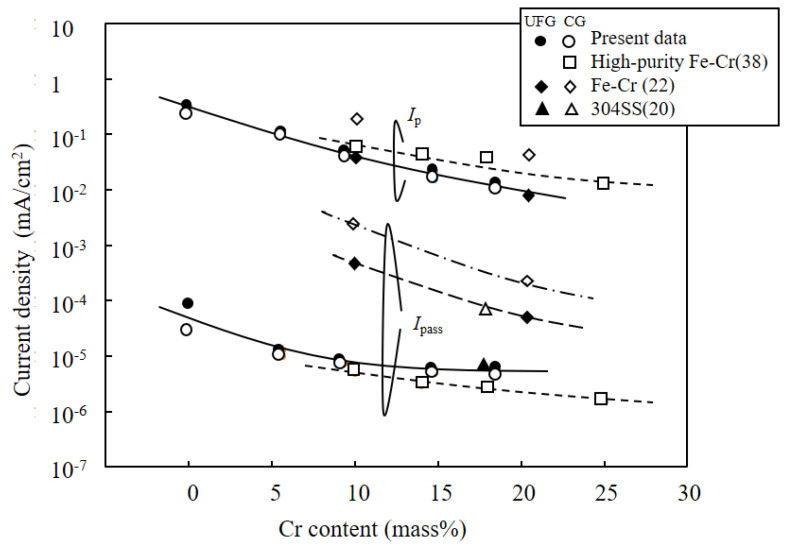
Peak current density *I*_c_ and passive current density *I*_pass_ of the dynamic polarization curves in 1.0 M H_2_SO_4_ solution (Figure 6) as a function of Cr content.

**Figure 8 materials-15-01007-f008:**
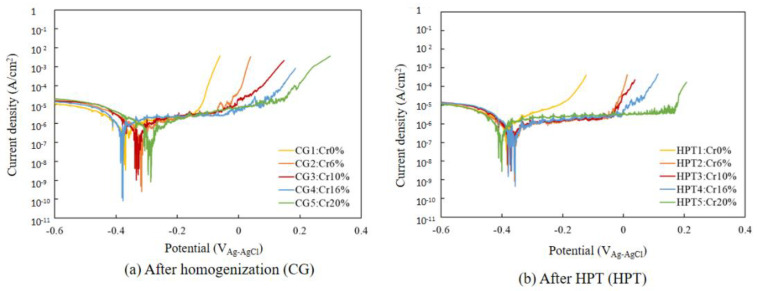
Dynamic polarization curves in 3.5 mass % NaCl solution, after (**a**) homogenization and (**b**) HPT processing.

**Figure 9 materials-15-01007-f009:**
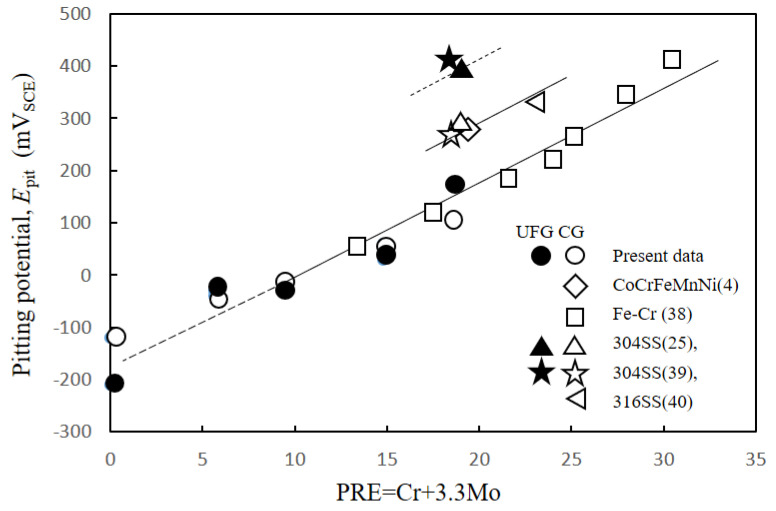
Critical pitting potential in 3.5 wt.% NaCl solution as a function of pitting resistance equivalence number (PRE = Cr + 3.3 Mo). Open and solid marks indicate CG and UFG specimens, respectively. Unit of the potential is converted to mV_SCE_ according to references.

**Figure 10 materials-15-01007-f010:**
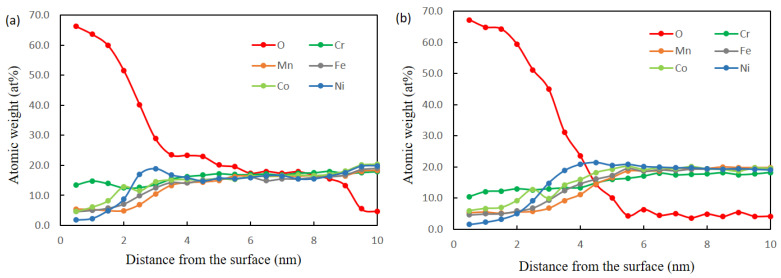
EDS depth profiles of the alloy elements in the passive films of 20 at% Cr specimen after passivation for 2 h at +700 mV_Ag/AgCl_: (**a**) homogenized (CG5) and (**b**) HPT process (HPT5).

**Figure 11 materials-15-01007-f011:**
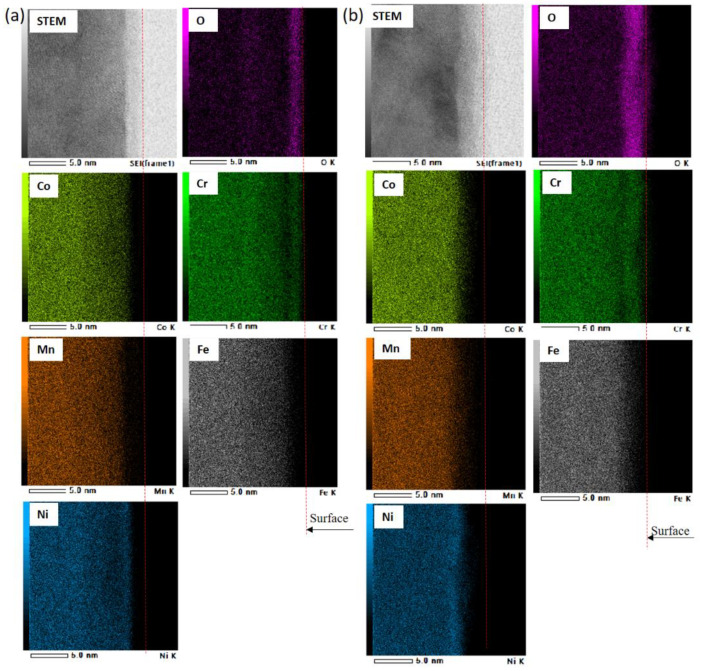
Cs-corrected STEM image and elemental maps of in the passive films of 20 at.% Cr specimen after passivation for 2 h at +700 mV_Ag/AgCl_: (**a**) homogenized (CG5) and (**b**) HPT process (HPT5).

**Table 1 materials-15-01007-t001:** Chemical composition of materials (at.%).

No.	Cr	Co	Fe	Ni	Mn
1	0	25	25	25	25
2	6	23.5	23.5	23.5	23.5
3	10	22.5	22.5	22.5	21
4	16	21	21	21	21
5	20	20	20	20	20

## Data Availability

The raw processed data required to reproduce these findings cannot be shared at this time as the data also form part of an ongoing study.

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
