# Peer review of "Corrosion Behavior of Ultrafine-Grained CoCrFeMnNi High-Entropy Alloys Fabricated by High-Pressure Torsion"

_materials, 2022, doi:10.3390/ma15031007_

Round 1

Reviewer 1 Report

In this study, nanocrystalline CoCrFeMnNi HEA alloys are prepared through SPD and corroded using aqueous 0.5 M H2SO4 and 3.5% NaCl solutions. Major conclusions of the work:

  1. Similar active-passive, passive, and transpassive behavior to Fe-Cr
  2. Increasing Cr resulted in increased passivation
  3. No difference in corrosion behavior with decreasing grain size in either solution

In this reviewer’s opinion, while the experimental work appears sound, the discussion in this work requires significant revision.

Minor comments:

  • Refrain from reporting atomic/mass percent (use counts) when measuring oxygen using STEM EDS. Surface oxidation of the lamella will significantly skew these values.
  • To facilitate reader comprehension, the colors in the EDS depth profiles in Figure 10 should match the maps in Figure 11.
  • English is overall fine. However, there are two bizarre sentences in line 351 in the conclusion: “They should start with a bullet. This is named conclusions.”
  • Grain size counting methodology should be mentioned (from XRD or TEM?).

The discussion is difficult to follow and at times contradictory. The authors acknowledge an increase in passivation and attribute it to enriched Cr in the passive films. However, in the discussion, the authors note the lack of selective dissolution reported in literature for HEAs which would suppress Cr enrichment. The authors then go on to claim that grain boundaries in HEAs may not provide fast diffusion channels. Understandable, however, the authors then end the discussion by stating: “grain refinement or high density of grain boundaries may facilitate Cr diffusion and local enrichment in the passivation film.” If this is the case, why doesn’t the SPD HEA demonstrate higher passivation?

Note that some prior studies have examined the corrosion behavior of HEA alloys and the effect of reducing grain size:

Han, Z.; Ren, W.; Yang, J.; Tian, A.; Du, Y.; Liu, G.; Wei, R.; Zhang, G.; Chen, Y. The corrosion behavior of ultra-fine grained CoNiFeCrMn high-entropy alloys. Journal of Alloys and Compounds 2020, 816, 152583. DOI: https://doi.org/10.1016/j.jallcom.2019.152583.

Wang, Y.; Jin, J.; Zhang, M.; Wang, X.; Gong, P.; Zhang, J.; Liu, J. Effect of the grain size on the corrosion behavior of CoCrFeMnNi HEAs in a 0.5 M H2SO4 solution. Journal of Alloys and Compounds 2021, 858, 157712. DOI: https://doi.org/10.1016/j.jallcom.2020.157712.

Both of the above papers demonstrate an appreciable reduction in corrosion behavior at the smallest grain sizes due to unstable passive films. The authors do not address any prior corrosion studies on UFG HEAs; how do the authors address this discrepancy? The authors could also address some of this work in the introduction.

Author Response

First of all, let us thank the reviewer for all valuable comments and suggestions. We responded as follows,

In this study, nanocrystalline CoCrFeMnNi HEA alloys are prepared through SPD and corroded using aqueous 0.5 M H2SO4 and 3.5% NaCl solutions. Major conclusions of the work:

  1. Similar active-passive, passive, and transpassive behavior to Fe-Cr
  2. Increasing Cr resulted in increased passivation
  3. No difference in corrosion behavior with decreasing grain size in either solution

In this reviewer’s opinion, while the experimental work appears sound, the discussion in this work requires significant revision.

It is found that the local enrichment of Cr elements in the passivation films was observed by STEM-EDS. Therefore, Cr plays key role in forming passivation films and enhancing corrosion resistance as in the well-known binary Fe-Cr alloys. Throughout the discussion, effect of grain refinement on corrosion behavior was discussed by comparing with Fe-Cr alloys. In the majority of past studies show that grain refinement smaller than 0.1mm enhances the corrosion resistance in Fe-Cr alloys. However, as shown in corrosion testing in H2SO4, effect of grain size reduction on passivation capability is very small. In our paper, we discuss why corrosion behavior is NOT enhanced by SPD in HEAs where Cr plays key role as in Fe-Cr alloys. It is considered that Cr diffusion and selective dissolution of Fe in the initial stage of passivation does NOT occurs in HEAs as in Fe-Cr alloys

 (1)Refrain from reporting atomic/mass percent (use counts) when measuring oxygen using STEM EDS. Surface oxidation of the lamella will significantly skew these values.

While the comments are understandable, the authors still consider the atomic percent is more suitable than counts (absolute value) by the following reason.

1)X-ray counts can be affected by the thickness of the sample, thus, X-ray counts of the passivation films becomes lower than the matrix. On the other hands, atomic fraction is NOT affected by the thickness or location.

2)X-ray peak of Ka and Kb of Cr, Mn, Fe, Co, Ni are superimposed, which make X-ray counts larger than the real fraction of elements.

3)The atomic fraction has more generality than the absolute value of X-ray counts, thus we can compare with the other papers.

  • To facilitate reader comprehension, the colors in the EDS depth profiles in Figure 10 should match the maps in Figure 11.

We modified Figure 10 accordingly.

  • English is overall fine. However, there are two bizarre sentences in line 351 in the conclusion: “They should start with a bullet. This is named conclusions.”

The sentences were deleted accordingly.

  • Grain size counting methodology should be mentioned (from XRD or TEM?).

Grain size was evaluated by TEM. The description is added accordingly.

  • The discussion is difficult to follow and at times contradictory. The authors acknowledge an increase in passivation and attribute it to enriched Cr in the passive films. However, in the discussion, the authors note the lack of selective dissolution reported in literature for HEAs which would suppress Cr enrichment. The authors then go on to claim that grain boundaries in HEAs may not provide fast diffusion channels. Understandable, however, the authors then end the discussion by stating: “grain refinement or high density of grain boundaries may facilitate Cr diffusion and local enrichment in the passivation film.” If this is the case, why doesn’t the SPD HEA demonstrate higher passivation?

I think this comment is related with the first comment, let me reiterate again as follows. The local enrichment of Cr elements in the passivation films suggests Cr plays key role in forming passivation films and enhancing corrosion resistance as in the well-known binary Fe-Cr alloys. In the majority of past studies show that grain refinement smaller than 0.1mm enhances the corrosion resistance in Fe-Cr alloys. In our paper, we discuss why corrosion behavior is NOT enhanced by SPD in HEAs whereas it is very much enhanced in Fe-Cr alloys. It is considered that Cr diffusion and selective dissolution of Fe in the initial stage of passivation does NOT occurs in HEAs as in Fe-Cr alloys.

  • Note that some prior studies have examined the corrosion behavior of HEA alloys and the effect of reducing grain size:

Han, Z.; Ren, W.; Yang, J.; Tian, A.; Du, Y.; Liu, G.; Wei, R.; Zhang, G.; Chen, Y. The corrosion behavior of ultra-fine grained CoNiFeCrMn high-entropy alloys. Journal of Alloys and Compounds 2020816, 152583. DOI: https://doi.org/10.1016/j.jallcom.2019.152583.

Wang, Y.; Jin, J.; Zhang, M.; Wang, X.; Gong, P.; Zhang, J.; Liu, J. Effect of the grain size on the corrosion behavior of CoCrFeMnNi HEAs in a 0.5 M H2SO4 solution. Journal of Alloys and Compounds 2021858, 157712. DOI: https://doi.org/10.1016/j.jallcom.2020.157712.

Both of the above papers demonstrate an appreciable reduction in corrosion behavior at the smallest grain sizes due to unstable passive films. The authors do not address any prior corrosion studies on UFG HEAs; how do the authors address this discrepancy? The authors could also address some of this work in the introduction.

Thank you so much valuable comments. We addressed these two papers in the Introduction. Canter alloys is considered to be single phase (FCC) structure. However, the complete homogeneous distribution of the five elements are very difficult to obtain, and some elements tend to be segregated during cooling after annealing.  Both studies controlled grain size by cold-rolling and subsequent annealing, and segregation of some elements can not be avoided. Indeed, Han et al reported that sigma-phase and Cr-depleted area was attributed to low corrosion resistance of UFG materials. This clouds the pure effect of grain size on corrosion behavior. On the other hands, SPD such high-pressure torsion can reduce grain size with no segregation. Even it can homogenize the elements by reshuffling of severe plastic deformation. Thus, we can expect to obtain “pure effect” of grain size reduction in nano range compared with rolling+annealing approach.

Reviewer 2 Report

This paper studied the effect of different Cr content on the corrosion resistance of CoCrFeMnNi alloy. Through the polarization curve and STEM observation in sulfuric acid solution, the higher the content of Cr, the better the corrosion resistance. The structure of the paper is clear, the logic is smooth, and it has very important practical application value, but some details need to be modified and polished:

  1. Align the image in the center. Including but not limited to: adjust the position of Figure 1, Figure 2 and Figure 3 to make the left and right gaps of the picture equal.
  2. In Figure 1, after HPT, the peak of (111) orientation is the highest. Could you explain the reasons for this?
  3. In line 183, the sample is in 0.5M H2SO4, which is contrary to the description of putting the sample in 1M H2SO4 during the experiment in line 94, please modify it.
  4. Please maintain the same font format in Figure 7. It is recommended to change the font to "Times New Roman".
  5. It is written in line 265 that Ni is enriched on the surface of CoCrFeMnNi alloy, but it is not found in STEM figure 11. Please explain the reason for the enrichment of Ni element.
  6. Some sentences in the text should not be divided into sections, including but not limited to: lines 193 and 199; Lines 221 and 228.

Author Response

First of all, let us thank you for all valuable comments and suggestions. We responded as follows,

  1. Align the image in the center. Including but not limited to: adjust the position of Figure 1, Figure 2 and Figure 3 to make the left and right gaps of the picture equal.

The figures were modified accordingly.

  1. In Figure 1, after HPT, the peak of (111) orientation is the highest. Could you explain the reasons for this?

We are not sure. HPT is gives quasi-commpressive state. Thus, slip plane of FCC phase become parallel to compressive plane. Texture in both samples are almost same, thus it does not influence the corrosion behavior.

  1. In line 183, the sample is in 0.5M H2SO4, which is contrary to the description of putting the sample in 1M H2SO4 during the experiment in line 94, please modify it.

The tests were done in 1 M H2SO4 solution. It was modified accordingly.

  1. Please maintain the same font format in Figure 7. It is recommended to change the font to "Times New Roman".

Figure 7 was modified accordingly.

  1. It is written in line 265 that Ni is enriched on the surface of CoCrFeMnNi alloy, but it is not found in STEM figure 11. Please explain the reason for the enrichment of Ni element.

Ni is enriched beneath the passivation film (not in the film). One can recognize the enrichment in the band of dense color next the passivation film

  1. Some sentences in the text should not be divided into sections, including but not limited to: lines 193 and 199; Lines 221 and 228.

They are modified accordingly.

Round 2

Reviewer 1 Report

Thank you to the author's for their response. The changes made by the authors are adequate and appropriate; at this stage, this paper is acceptable for publication in Materials.